# Label-Free Sensing of Cell Viability Using a Low-Cost Impedance Cytometry Device

**DOI:** 10.3390/mi14020407

**Published:** 2023-02-09

**Authors:** Bowen Yang, Chao Wang, Xinyi Liang, Jinchao Li, Shanshan Li, Jie Jayne Wu, Tanbin Su, Junwei Li

**Affiliations:** 1Hebei Key Laboratory of Smart Sensing and Human-Robot Interactions, School of Mechanical Engineering, Hebei University of Technology, Tianjin 300130, China; 2Institute of Biophysics, School of Health Sciences and Biomedical Engineering, Hebei University of Technology, Tianjin 300401, China; 3State Key Laboratory of Reliability and Intelligence of Electrical Equipment, Hebei University of Technology, Tianjin 300132, China; 4Department of Electrical Engineering and Computer Science, The University of Tennessee, Knoxville, TN 37919, USA

**Keywords:** microfluidic impedance cytometry, low-cost ITO electrodes, cell viability analysis, label-free sensing, sensitive electrodes

## Abstract

Cell viability is an essential physiological status for drug screening. While cell staining is a conventional cell viability analysis method, dye staining is usually cytotoxic. Alternatively, impedance cytometry provides a straightforward and label-free sensing approach for the assessment of cell viability. A key element of impedance cytometry is its sensing electrodes. Most state-of-the-art electrodes are made of expensive metals, microfabricated by lithography, with a typical size of ten microns. In this work, we proposed a low-cost microfluidic impedance cytometry device with 100-micron wide indium tin oxide (ITO) electrodes to achieve a comparable performance to the 10-micron wide Au electrodes. The effectiveness was experimentally verified as 7 μm beads can be distinguished from 10 μm beads. To the best of our knowledge, this is the lowest geometry ratio of the target to the sensing unit in the impedance cytometry technology. Furthermore, a cell viability test was performed on MCF-7 cells. The proposed double differential impedance cytometry device has successfully differentiated the living and dead MCF-7 cells with a throughput of ~1000 cells/s. The label-free and low-cost, high-throughput impedance cytometry could benefit drug screening, fundamental biological research and other biomedical applications.

## 1. Introduction

Cell viability analysis is of great significance in drug screening. Traditionally, cell viability can be determined by staining cells with Trypan Blue (TB) and fluorescent indicators [1,2,3]. TB is a commonly used dye to label dead cells that can pass through the permeabilized membranes of dead cells to stain the nucleus blue, thus distinguishing living and dead cells under the optical microscope [4]. However, the staining results of TB are not very accurate. Previous studies have shown that when the cell viability is below 70%, the results may be over-estimated [5]. Another fluorescent indicator used to stain living cells is Calcein AM, which can hydrolyze through the cell membrane and the endogenous esterase in living cells to produce a highly negatively charged polar molecule, Calcein, which cannot penetrate the cell membrane, thus remaining in cells. Calcein can emit strong green fluorescence. Dead cells lack esterase or have a very low esterase activity, so all dead cells will not have a green fluorescence [6]. Nonetheless, cell staining usually requires invading the interior of a cell, and the fluorescent indicator may be toxic [7]. Furthermore, while fluorescence flow cytometry is a conventional method for measuring cell viability, it requires expensive and bulky equipment, as well as staining cells [8]. 

In recent years, with the development of microfluidics, the microfluidic impedance cytometer (MIC) based on Coulter counter-conception has attracted extensive attention from researchers [9,10,11]. Compared with a conventional flow cytometer, the MIC is non-invasive, portable, and label-free, and has been widely used in the detection of cells [12,13], bacteria [14,15], and other bioparticles [16]. Sensing electrodes are an important part of MICs. The detecting electrodes generate an electric field to form a detection region, and particles passing through the detection region will cause a variation in the electric field to generate impedance signals [17]. The MIC has many applications in the field of flow cytometry, such as in investigating the dielectric properties of white blood cells (WBCs) and MCF-7 cancer cells [18], differentiating cell death and apoptosis morphology [19], and blood cell counting [20]. The MIC device in this study provides an efficient and accurate platform for cell viability analysis.

The detection electrodes in an MIC are generally made of Au, which is expensive [21]. Further, the detection electrodes often need to be fabricated with a critical dimension of around 20-micron, which requires relatively high fabrication accuracy. Consequently, the production cost and difficulty of the detection electrodes increase significantly [22], especially for massive production in real applications. Therefore, narrow Au electrodes are not viable for low-cost disposable detection cases. To solve this problem, many low-cost methods to produce electrodes have been reported. For example, Tang et al. [23] used liquid electrodes in their cytometer instead of Au electrodes to avoid the complex steps involved in producing microfabricated metal electrodes. However, the accuracy and sensitivity of the liquid electrodes were poor, and the conductive solution of the liquid electrodes was suspected to potentially contaminate sample solutions. Recently, Cheng et al. [24] used molten liquid metal for the 3D-detection electrodes in their device by using a Sn42Bi58 alloy solder wire instead of conventional electrode materials (e.g., Au and Cr). Instead of making conventional electrodes, they melted the Sn42Bi58 alloy solder wire into metal liquid and injected the melted alloy into the microelectrode layer at a controlled driving pressure. Although this method provides a simple and low-cost method for electrode fabrication, the accuracy of the electrode shapes was relatively low. As printed circuit boards (PCBs) have become popular all over the world, electrodes on PCBs are cheap and can be manufactured in large quantities. Guo et al. [25] reported a cheap copper electrode integrated on a PCB and stacked the polydimethylsiloxane (PDMS) microchannel on the PCB to detect circulating tumor cells. However, the electrodes on the PCB are always separated from the microchannel by a layer of PDMS or glass, which weakens the electric field in the channel and reduces the sensitivity of the electrodes [26,27]. In this study, we proposed a low-cost microfluidic impedance cytometry device with 100-micron scale indium tin oxide (ITO) electrodes to achieve a level of performance comparable to the 10-micron scale Au electrodes.

The relatively large sizes of ITO electrodes reduce the requirements for fabricating accuracy and lower the cost. To improve the measurement accuracy, we used a seven-electrode coplanar configuration as the detection region. This electrode configuration is referred to as a double differential configuration. Previous work has verified that the double differential configuration has a higher sensitivity and can eliminate the position dependence of particles. The electric diameter after calibration has better performance [28]. We also built a 3D finite element method (FEM) of our MIC to compare the signals generated by different sizes of particles passing through the detection region. The results showed that our device has a minimum resolution size of around 5 μm particle. Then we tested our device’s sensitivity by using polystyrene beads and MCF-7 cells. The experiment result showed that our device can distinguish the 7 μm beads from the 10 μm beads. Thus, we have further demonstrated that the living and dead MCF-7 cells can be separated with an accuracy of 94.5% with our device. Our MIC device can discriminate between dead and living cells by processing the cell impedance signal. It is also non-invasive, label-free, and has a very high accuracy. Previous studies have demonstrated that 20 μm Au electrodes can distinguish living MCF-7 cells from dead MCF-7 cells [11]. In this work, we succeeded in distinguishing living and dead MCF-7 cells using 100 μm ITO electrodes. The typical differential current signals from the Au electrodes and ITO electrodes are provided in Appendix A. Our device, with a lower production cost (less than USD 1 per chip; a cost breakdown table is shown in Appendix A) and a high sensitivity, has broad application prospects in biology tests and rapid detection technology.

## 2. Materials and Methods

### 2.1. Working Principles

The MIC can detect a single cell in real-time. When the suspension of cells passes through the detection region, it will generate complex impedance signals that can indicate the dielectric properties of cells, discriminate between living and dead cells, detect cell apoptosis, etc. [19]. The cell’s electrical impedance is defined as the ratio of the voltage to the current. The impedance is a measure of the dielectric properties (permittivity and conductivity) of the system [29]. The cell electrical impedance equation is given by
(1)$$\tilde{Z}=\frac{\tilde{V}}{\tilde{I}}$$
where $\tilde{Z}$ is the cell electrical impedance, $\tilde{V}$ is the voltage, $\tilde{I}$ is the current, and the superscript ‘’~” represents the complex number.

The complex permittivity of a mixture of cells in a suspension is usually described by Maxwell’s mixture theory. The single-shelled spherical model has been widely used, as shown in Figure 1a [30]. Based on Maxwell’s mixture theory, the single-shelled spherical model is composed of a conducting sphere (cytoplasm) and an insulating thin shell (cell membrane).

The complex permittivity of the mixture is ${\tilde{\varepsilon }}_{\mathrm{mix}}$:(2)$${\tilde{\varepsilon }}_{\mathrm{mix}}={\tilde{\varepsilon }}_{\mathrm{med}}\frac{1+2\Phi {\tilde{f}}_{\mathrm{CM}}}{1-\Phi {\tilde{f}}_{\mathrm{CM}}}$$
(3)$${\tilde{f}}_{\mathrm{CM}}=\frac{{\tilde{\varepsilon }}_{c}-{\tilde{\varepsilon }}_{\mathrm{med}}}{{\tilde{\varepsilon }}_{c}+2{\tilde{\varepsilon }}_{\mathrm{med}}}$$
where ${\tilde{f}}_{\mathrm{CM}}$ is the Clausius–Mossotti factor, Φ is the volume fraction. The subscripts “*c*” and “*med*” refer to cell and medium, respectively. The complex dielectric parameters of cell ${\tilde{\varepsilon }}_{c}$ can be expressed by the following equation:(4)$${\tilde{\varepsilon }}_{c}={\tilde{\varepsilon }}_{\mathrm{mem}}\frac{\gamma^{3}+2\left(\frac{{\tilde{\varepsilon }}_{i}-{\tilde{\varepsilon }}_{\mathrm{mem}}}{{\tilde{\varepsilon }}_{i}+2{\tilde{\varepsilon }}_{\mathrm{mem}}}\right)}{\gamma^{3}-\left(\frac{{\tilde{\varepsilon }}_{i}-{\tilde{\varepsilon }}_{mem}}{{\tilde{\varepsilon }}_{i}+2{\tilde{\varepsilon }}_{mem}}\right)}$$
(5)$$\gamma =\frac{R+d}{R}$$
where the subscripts “*mem*” and “*i*” represent the cell membrane and cytoplasm, respectively. *R* and *d* represent the internal radius of the cell and the thickness of the cell membrane, respectively. The impedance of a system (${\tilde{Z}}_{mix}$) is expressed by the following equation:(6)$${\tilde{Z}}_{\mathrm{mix}}=\frac{1}{j\omega {\tilde{\varepsilon }}_{mix}G_{f}}$$
where $G_{f}$ is a geometric constant, generally the ratio of electrode area to electrode gap. $j=\sqrt{-1}$, ω is the electric field frequency.

### 2.2. Finite Element Modeling

A 3D finite element model was established to simulate the current density within the microchannel and the real-time differential current signal changes when particles or cells pass through the microchannel. We used the AC/DC Module of COMSOL Multiphysics 5.6 (COMSOL AB, Kgs. Lyngby, Denmark) for the impedance cytometry simulation. Our finite element model used the current conservation equation based on Ohm’s law from the AC/DC modules:(7)$$J=\left(\sigma +j\omega \varepsilon_{0}\varepsilon_{r}\right)\nabla U$$
where σ is conductivity, $\varepsilon_{0}$ and $\varepsilon_{r}$ is the vacuum dielectric constant and relative dielectric constant, respectively, and ∇U is the potential difference.

The exterior boundary conditions of this model were set to ‘Electric Insulation’, the interior boundaries between different sub-domains were set to ‘Continuity’, and the initial potential value of all domains was 0 V. The central electrode was set as ‘Terminal’, and the voltage amplitude was set to 1 V at 0° phase angle. It had two neighboring electrodes set to the ground (GND) electrodes. Two extra electrodes on both sides of the detection region were also set as ‘Terminal’, which had the same voltage amplitude but opposite phase (180° phase angle). The floating electrodes used for particle size calibration were located between the ‘Terminal’ electrodes and the GND electrodes. The floating electrodes were set to ‘Floating Potential’. To simulate the flow of particles in the microchannel, we defined the flow of particles as the change of materials inside the microchannel, rather than creating a solid particle [31]. The initial coordinates of particles were set to $x_{0}$, $y_{0}$, and $z_{0}$, and parameter scanning $x_{0}$ was used to simulate the flow of particles in the microchannel. A variable DFC was defined to calculate the distance to the particle or cell center:(8)$$\mathrm{DFC}=\left({\left(x-x_{0}\right)}^{2}+{\left(y-y_{0}\right)}^{2}+{\left(z-z_{0}\right)}^{2}\right)$$

The material inside the microchannel is defined as:(9)$$\mathrm{Electrical}\;\mathrm{conductivity}=\mathrm{sigma}\_\mathrm{par}+\left(\mathrm{sigma}\_\mathrm{sol}-\mathrm{sigma}\_\mathrm{par}\right)*(\mathrm{DFC}>r_{0}^{2})$$
(10)$$\mathrm{Relative}\;\mathrm{permittivity}=\mathrm{eps}\_r\_\mathrm{par}+\left(\mathrm{eps}\_r\_\mathrm{sol}-\mathrm{eps}\_r\_\mathrm{par}\right)*(\mathrm{DFC}>r_{0}^{2})$$
where sigma_par, eps_r_par, sigma_sol, eps_r_sol represent conductivity and relative permittivity of particles and medium, respectively, and $r_{0}$ represents particle radius. The expression $\mathrm{DFC}>r_{0}^{2}$ is a Boolean expression. When the expression is true, the expression is set to 1 outside the particle and 0 inside the particle for false. The equations can solve the internal and external material of particles in the microchannel simultaneously [31].

### 2.3. Microfluidic Chip Fabrication and Measurement Setup

The MIC chip mainly contains two components: a PDMS microchannel and ITO electrodes. The width of the microchannel is 100 μm, and the channel height is 30 μm. The detection region has the dimensions of 1260 μm × 100 μm × 30 μm. Particles and cells can achieve high throughput without blocking the microchannel at this size. The fabrication of the PDMS microchannel was shown in Figure 1b. First, (i) prepare a 4-inch silicon wafer. (ii) Spin the SU–8 photoresist (MicroChem, Westborough, MA, USA) onto the cleaned silicon wafer at a speed of 3500 rpm by a spin coater (TB616 Spin Coater, Sysile) to form a negative photoresist with a thickness of 30 μm on the silicon wafer. The mask pattern on the silicon chip was pre-designed with Auto CAD software. (iii) After pre-baking, UV exposure, and post-baking, wait for 5 min and let the photoresist develop at room temperature. The silicon wafer containing the designed photoresist pattern was used as the mold for the MIC channels. (iv) Prepare 40 g PDMS (SYLGARD 184, Dow Corning) and 4 g curing agent. After mixing, use a vacuum pump to remove the air bubbles. Then pour the mixture onto the SU-8 mold and heat it at 80 °C for 60 min. (v) The solidified PDMS was carefully removed from the mold, the microchannel was peeled off with a knife, and 0.7 mm diameter holes were punched at the inlet and outlet of the PDMS model. (vi) The solidified PDMS and the glass with ITO electrodes were plasma cleaned and well bonded using a plasma machine (PDC-MG, PTL Technology Co, Ltd., Shenzhen China). Finally, the bonded MIC chip was placed on the heating plate for 30 min at 80 °C) [32]. The schematic diagram of the bonded MIC chip is shown in Figure 1c.

We have also simplified the fabrication of the ITO electrodes. The traditional method of fabricating ITO electrodes is by photolithography as shown in Figure 2a. Our ITO electrode pattern was fabricated with a 355 nm UV laser cutting machine (MVU12, Wuhan Hero Optoelectronics Technology Co., Ltd., Wuhan, China), as shown in Figure 2b. This is an efficient and cost-effective method compared to photolithography. This fabrication method enables the mass production of ITO electrodes.

Figure 3 shows the configuration of the double differential microfluidic impedance cytometry system. Particle or cell suspensions are passed through the detection region by the pressure pump (OB1 MK3+, Elveflow, Paris, France). The impedance spectrometer (HF2IS, Zurich Instrument, Zurich, Switzerland) provides the exciting signals at ±1 V AC voltage with a phase angle of 0° and 180°, respectively. As shown in Figure 2, 1 V AC voltage with a phase angle of 0° was applied to the central electrode, and a voltage with a 180° phase angle was applied to two side electrodes. Two GND electrodes were connected to a current amplifier (HF2TA, 100 dB gain), and the sampling frequency was set to 899 Hz. The impedance spectrometer was connected to a computer to process the differential current signal. 

When the particles or cells entered the microchannel, we used the inverted microscope (Nikon Eclipse TI-S, Tokyo, Japan) equipped with a CCD camera (Nikon DS-QI 2) to take optical observations. At the beginning, 7 μm and 10 μm polystyrene beads were injected into the MIC chip to verify the accuracy of our device, and then the cell suspensions were pumped into the channel. Here, we used living and dead MCF-7 cells for cell viability analysis. A custom-built Python script was used to process the data and extract the electrical signal of particles or cells, including the position calibration factor and the original electric diameter, and to obtain the calibrated electrical diameter through the linear fitting algorithm [33], as described in the Appendix A.

### 2.4. Sample Preparation

To evaluate the sensitivity of our MIC device, we prepared 7 μm and 10 μm polystyrene beads. The beads were diluted in tubes containing a photographic buffered saline (PBS) with a concentration of around $3\times 10^{5}$ particles per mL. The conductivity of the medium measured by the conductivity instrument (DDS-307A, Shanghai Rex Instruments, Shanghai, China) was 1.6 S $m^{-1}$. Before the beads were loaded into the impedance flow cytometry device, the 7 μm (10 μL, 3 × 10^5^ beads/mL) and 10 μm (10 μL, 3 × 10^5^ beads/mL) beads were mixed well.

As shown in Figure 4, we also prepared the living and dead MCF-7 cells to verify that our device can discriminate between living and dead cells. The MCF-7 breast cancer cells were observed under a microscope, and when they had grown to 70% to 80% in a cell culture flask (BD Biosciences), the passage culture operation could be performed. The cell passage culture operation is shown in Figure 4a. The supernatant was discarded from the cell culture bottle, and a pancreatic enzyme digestion solution was added and left to stand. When the cells fell down, the PBS medium was added to stop digestion, and the mixture was then repeated with a sterile dropper. The mixture was placed in a 15 mL centrifuge tube and centrifuged at 1000 r/min for 5 min, with the supernatant discarded after centrifugation. The medium was added into the tube for full mixing, and finally, the cell suspension was inoculated into the cell culture bottle. The cell culture flask was placed in a cell incubator (Forma 381, Thermo Fisher Scientific, Waltham, MA, USA) at 37 °C with 5% CO_2_ for further culture. Before injection into our device, the MCF-7 cells were trypsinized and resuspended in DPBS (Thermo Fisher Scientific) with a concentration of 0.2 wt% polyethylene oxide (PEO, $M_{W}$ = 600 kDa, Sigma-Aldrich, St. Louis, MO, USA) for consistent cell alignment and ordering in the impedance sensing channel [19]. Additional steps are required for dead cell preparation. The MCF-7 cells were cultured in Dulbecco’s Modified Eagle’s Medium (DMEM, Thermo Fisher Scientific) supplemented with 10% fetal bovine serum (FBS, Thermo Fisher Scientific) and in a cell incubator at 37 °C and 5% carbon dioxide. To induce necrosis, after overnight incubation in a cell incubator, the cells were exposed to heat shock through incubation at 60 °C for 30 min. Then, the cells were washed 3 times in the PBS and centrifuge, as shown in Figure 4b. When the live and dead cells are well prepared, a suspension containing both live and dead cells were obtained to mimic the mixture of cells.

## 3. Results and Discussion

### 3.1. Differential Current Signal by FEM

In Section 2.2, we described the process for establishing the finite element method of microfluidic impedance flow cytometry. In this section, we will discuss the process of finite element calculation and the simulation of differential current signals. The geometry and physical settings were kept the same as the experimental system. In this model, free quad meshes were employed as domain elements due to their excellent shape adaptation characteristics. Distribution meshing (number of elements was set at 20) was used in the electrodes area to achieve an accurate calculation of the current. The maximum element size of the meshes was set to 8 μm and the minimum element size was set to 0.01 μm. The floating electrode size was set to 80 μm. All other electrodes were set to 100 μm. The gap between the electrodes was set to 100 μm.

Electric field strength is critical to the sensitivity of the MIC detection of particles. The current density can reflect the magnitude of the electric field strength. From the current density distribution diagram as shown in Figure 5a, we can also see that the current density is relatively low at the top of the channel away from the electrodes, while the current density is relatively high at the bottom of the channel near the electrode. The current density also gets higher when it is close to the central electrode. This also explains why the signal peak at the center electrode is higher. After simulating the signals generated by particles of different sizes passing through the detection region, we set the particles’ diameters to 5 μm, 7 μm, 10 μm, 12 μm, and 15 μm. The differential current signals generated by different sizes of particles passing through the detection region are shown in Figure 5b. It can be seen that the amplitude of the signal increases with the particle diameter. Meanwhile, with the increase in particle diameter, the pulse width of the signal will also get bigger, which indicates that the particles may need more travel time to pass through the detection region. When the particle diameter is 5 μm, the differential signal is very close to the base line. The result showed that our device has a minimum resolution size of about 5 μm particle. The ratio between the cells/beads size and the electrode width is around 1:20. To the best of our knowledge, this is the lowest geometry ratio of the target to the sensing unit in the impedance cytometry technology. Such a low ratio allows reduced device fabrication difficulty and cost.

### 3.2. Particle Detection

To test the sensitivity of our MIC device, we used a pressure pump to inject the 1:1 mixture of 7 μm and 10 μm beads into our device. A 1 V AC voltage was applied to the center electrode at a frequency of 500 kHz, and −1 V AC voltages were applied to the left and right side electrodes at a frequency of 500 kHz. The beads passed through the detection region and generated a differential current signal, which is shown in Figure 6a. As reflected in Figure 5a, the signal generated by the beads is obviously different from the noise, and its signal-to-noise ratio (SNR) reaches 23.45 dB. The signal amplification diagram of a single bead passing through the detection region is shown in Figure 6b. Our device can perfectly generate a double differential current signal, and as the signal-to-noise ratio is high, it can accurately extract the characteristics of the signal. Details of the data analysis of the double differential current signal are described in the Appendix A. Figure 6c shows the histogram of the beads with an electric diameter after calibration and the Gaussian fitting curves of the beads. It can be seen that the 7 μm and 10 μm beads can be discriminate and counted clearly. Figure 6d shows the density map of the beads. Drom different clusters in the figure, we can discriminate between the 7 μm and 10 μm beads. The coefficient of variation (CV) for the 7 μm and 10 μm beads measured with our device are 10% and 12.1%, respectively.

### 3.3. MCF-7 Cell Viability Profiling

To verify the biological application of our device, living and dead MCF-7 cells were prepared for testing. In the single-shell spherical model, the cell membrane is equivalent to a dielectric insulator. Cell membranes of dead cells are permeable because the membrane protein function and membrane integrity of dead cells are lost [34]. In addition, the composition of the cytoplasm will also change because the conductive medium can freely diffuse into dead cells and release intercellular contents. As a result, the dielectric properties of living cells and dead cells are different [19]. According to a previous study [29], electrical signals cannot penetrate the cell membrane below 1 MHz, and the information expressed by the impedance signal is the cell size. At a frequency above 1 MHz, the electrical signal can penetrate the cell membrane and express the information inside the cell. Opacity is defined as the ratio of high-frequency impedance amplitude to low-frequency impedance amplitude, which can be used to indicate the dielectric properties of cells, regardless of cell size changes [17]. Cheung et al. [10] proved that the opacity of red blood cells (RBCs) is different between living and dead cells. In this experiment, we used opacity as an indicator to test the dielectric properties of living and dead MCF-7 cells.

We used a pressure pump to inject the mixed suspensions of living and dead MCF-7 cells into the microchannel at a pressure of 10 mbar. The central electrode was applied with a voltage of 1 V, and the left and right side electrodes were applied with a voltage of −1 V. The AC voltage frequency was set to 500 kHz and 10 MHz, respectively, and the signal caused by the MCF-7 cells passing through the detection region is shown in Figure 7a. It can be seen from Figure 7a that the signal is distinct from noise, which can be used for cell counting and viability analysis. Figure 7b is the histogram of cell count, and the horizontal axis is the calibrated electric diameter. The living cells are larger than the dead ones, which is probably due to cell collapse after death. Moreover, the standard deviation of living cells is relatively larger than that of dead cells, which may be because the cell sizes of living cells are uneven in nature. After cell death, the membrane permeability changes, and the cell sizes tend to be the same. Figure 7c shows the density map of the MCF-7 cells. The horizontal axis of the density map is the calibrated electric diameter, and the vertical axis is the opacity. It can be seen that living cells could be distinguished from dead cells using the electrical diameter. Due to the difference in dielectric properties between the living and dead cells, their opacity also shows different behaviors. The living cells could be well fitted using both the Binomial distribution and Normal distribution models, with a mean value of 0.04219. However, the Normal distribution model fails to fit the opacity of the dead cells or mixed cells. Thus, we could distinguish the cell viability by the coefficient of variation (CV = SD/mu for Normal distribution). The CV of the living cells, the dead cells and the mixture is 26%, 178%, and 213% respectively. Thus, our device offers a novel strategy for cell viability evaluation for the MCF-7 cells. Our MIC is a label-free and low-cost strategy compared to the conventional flow cytometer.

### 3.4. Advantages and Limitations

This work presents an impedance cytometry system using a 100 μm wide ITO electrode, which is easy to fabricate at a low cost. Previous works have tended to adopt ten-micron scale gold electrodes as the detection electrodes, which are costly. Our larger ITO electrodes could achieve a comparable performance as the 10-micron scale Au electrodes. A comparison of this work and existing technologies is given in Appendix A. Moreover, we have tested the repeatability of our device using the same device, using different devices, and by different operators. With a correlation coefficient of 0.952, the experimental results show a strong consistency.

Our device also has some limitations. For example, it can only identify dead cells from living cells or distinguish different cells or beads with electrical characteristics. To achieve cell screening, we may need to integrate another unit to separate the living and dead cells. Adding a negative pressure pump or an electric field unit to collect living cells at the downstream for further analysis is suggested. The key technology is to define the triggering signal, which is the threshold to identify living cells. Thus, more work should be taken to improve the screening capability of our device.

## 4. Conclusions

In this work, we reported a low-cost MIC device that integrates large size ITO electrodes and a PDMS microchannel. The ITO electrodes are fabricated by wet-etching the ITO (Indium Tin Oxides) coated on borosilicate glass. This method can mass-produce ITO electrodes. Due to the electric field generated by coplanar electrodes being non-homogeneous, particles flowing through the channel with different trajectories will generate different signals. The double differential measure was used to eliminate the position dependence of particles and increase the sensitivity of our device. The electric diameter after calibration has shown a significant enhancement in our device’s accuracy for particle and cell detection. The detection electrodes of traditional impedance cytometry are made of Au, which is expensive and labor-intensive to produce. Our MIC device uses ITO electrodes with a 100-micron width, which is cheap and can reduce the difficulty in fabrication. To our knowledge, this is the lowest geometry ratio of the target to the sensitive unit in the impedance cytometry technology. We have demonstrated the sensitivity of our MIC device. First, we explored two different-sized beads, mixed them, and injected them into the MIC device. The results show that our MIC device can distinguish between the 7 μm and 10 μm beads. After testing the MCF-7 cells, the results showed our low-cost MIC device is able to characterize and discriminate between living and dead MCF-7 cells, which previously could only be discriminated by fluorescence staining in flow cytometry. The diameter of the cells can also be estimated from the calibrated electrical diameter obtained by our calculation. In conclusion, our device provides a simple and low-cost fabrication process and is promising for drug screening and biomedical applications.

## Figures and Tables

**Figure 1 micromachines-14-00407-f001:**
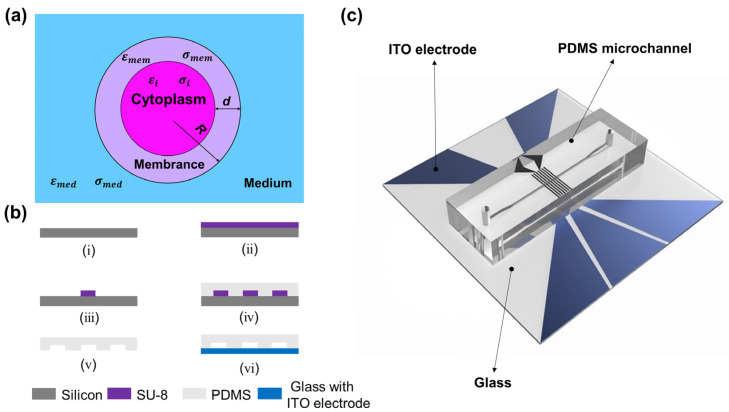
(**a**) Single-shell spherical model represents a single cell in suspension. ${\tilde{\varepsilon }}_{med}$, ${\tilde{\varepsilon }}_{mem}$ and ${\tilde{\varepsilon }}_{i}$ represent the complex permittivity of the suspending medium, cell membrane and cytoplasm, respectively; $\sigma_{med}$, $\sigma_{mem}$ and $\sigma_{i}$ represent the conductivity of the medium and cell membrane cytoplasm, respectively. (**b**) The process of fabricating an MIC chip: (i) preparing a clean silicon wafer, (ii) negative photoresist spin coating, (iii) photolithography, (iv) PDMS pouring, (v) demoulding, (vi) bonding. (**c**) Schematic of the MIC chip.

**Figure 2 micromachines-14-00407-f002:**
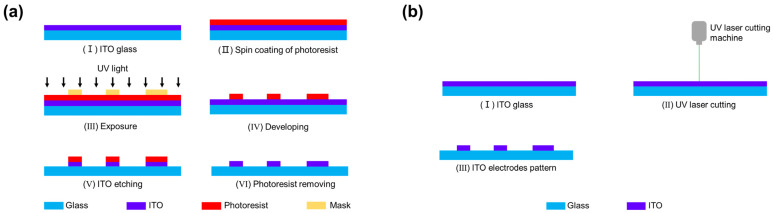
(**a**) Diagram of the photolithography microfabrication process for the ITO electrodes. (**b**) Diagram of the UV laser cutting fabrication process for the ITO electrodes.

**Figure 3 micromachines-14-00407-f003:**
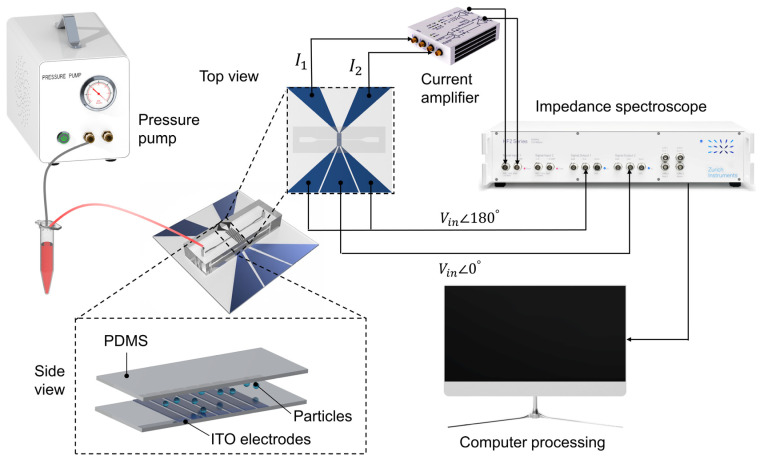
Schematic of the double differential microfluidic impedance cytometer setup consisting of a PDMS microchannel and ITO electrodes. A pressure pump injected particles into the PDMS microchannel. AC voltage signals generated by impedance spectroscope were applied to the detecting electrodes. A current amplifier was used to obtain the differential current signal, which is fed to the impedance spectroscope. Eventually, the signal is transmitted to a computer for processing.

**Figure 4 micromachines-14-00407-f004:**
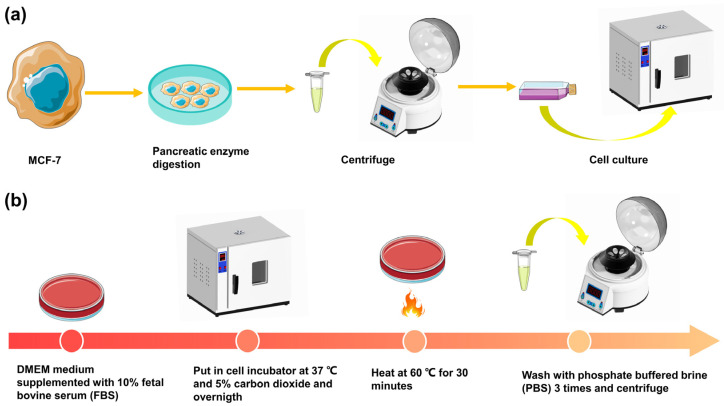
(**a**) The passage culture process for the MCF-7 cells. When the cells were attached to the wall and cell passaging could be performed, the old culture medium in the culture bottle was poured out, and the trypsin digestion solution was added for digestion. The cells were then centrifuged with a trypsin digestive juice mixture. After centrifugation, the supernatant was discarded and a new medium was added to the cell culture flask. The culture was carried out in a cell incubator. (**b**) Treatment of dead cells. First, the cultured live cells were removed from the cell incubator and heated at 60 °C for 30 min. Then we washed the dead cells 3 times with PBS before centrifugation. The dead MCF-7 cells were thus obtained.

**Figure 5 micromachines-14-00407-f005:**
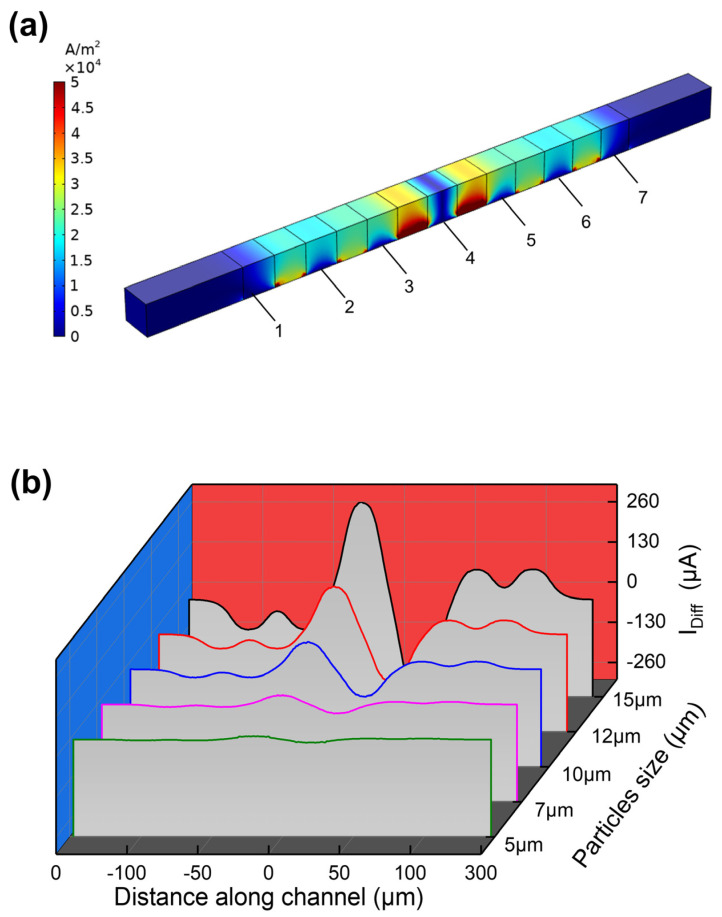
(**a**) Current density distribution. In the figure, number 4 represents the central electrode, numbers 3 and 5 represent the GND electrodes, numbers 2 and 6 represent the floating electrodes, and 1 and 7 represent the electrodes with opposite phase angles to the central electrode. (**b**) Differential current signals for different particle sizes: green represents 5 μm particles, pink represents 7 μm particles, blue represents 10 μm particles, red represents 12 μm particles, and black represents 15 μm particles.

**Figure 6 micromachines-14-00407-f006:**
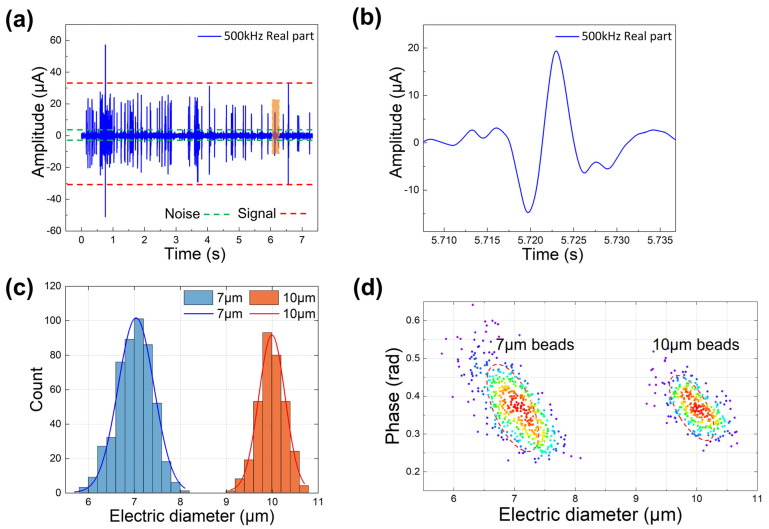
(**a**) Differential current signal for the 7 μm and 10 μm beads under the 1V AC voltage with a 500kHz frequency. The section between the dotted red lines represents the signal and the section between the dotted green lines represents the noise. (**b**) A differential current signal of a single particle passes through the detection region. (**c**) Histograms and Gaussian fitting curves of the electric diameter of 7 μm and 10 μm beads after calibration. (**d**) Density map of the electric diameter of the 7 μm and 10 μm beads after calibration.

**Figure 7 micromachines-14-00407-f007:**
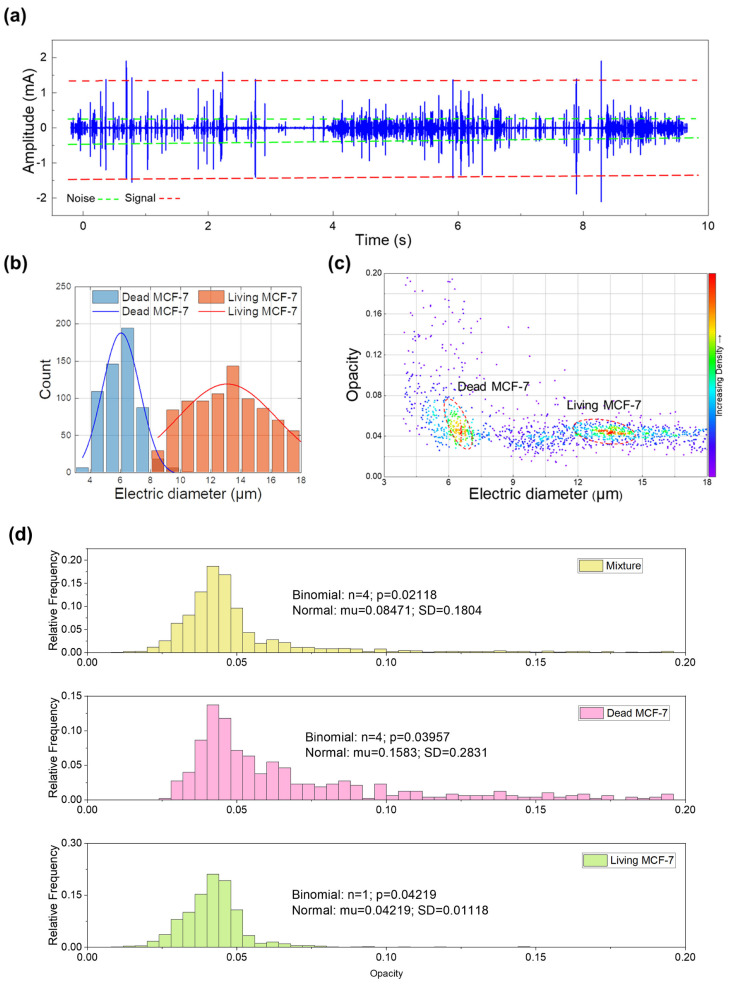
(**a**) The differential current signal caused by the living and dead MCF-7 cells passing through the detection region. The section between the dotted red lines represents signal and the section between the dotted green lines represents noise. (**b**) Histograms of the electric diameter of the living and dead MCF-7 after calibration. The blue and orange curves represent the Gaussian fitting curves. (**c**) Density map of the electric diameter of living and dead MCF-7 after calibration. (**d**) The statistical analysis of the opacity of living and dead cells.

## Data Availability

The data that support the findings of this study are available from the corresponding author upon reasonable request.

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
