# Peer review of "Label-Free Sensing of Cell Viability Using a Low-Cost Impedance Cytometry Device"

_micromachines, 2023, doi:10.3390/mi14020407_

Round 1

Reviewer 1 Report

The authors designed and built a microfluidic impedance cytometer for measuring the size and other properties of small particles in a suspension. As compared to previous approaches, relatively large (100 micron) ITO electrodes were used to produce a double differential current signal when a particle floats through the electrode zone. No fluorescence labelling or staining of the particles is required. The authors could successfully discriminate between polystyrene beads of sizes of 7 and 10 microns and between living and dead cancer cells. The experiments were supplemented by finite-element simulations. In my opinion, this is a very interesting paper in the field of particle or cell characterization, and it should be published after the following points have been addressed.

1. Equation (1) and below: The authors use sometimes an asterisk (e.g., Z*) and sometimes a tilde (e.g., epsilon~) to indicate complex quantities. I recommend using only one of these labels.Then, Z_mix in Eq. (6) and J in Eq. (7) are also complex and should have this label. Also epsilon_med, epsilon_mem, and epsilon_i in the caption of Fig. 1 are complex and should have the label.

2. Equation (10) and line below: The equation contains the variables eps_p_par and eps_p_sol, which are called eps_r_par and eps_r_sol in line 158. Consistent names should be used.

3. Line 194: The gain of the current amplifier is given as 100 kOhm. The gain of an amplifier is a dimensionless number, or it can be given in dB, if the logarithmic scale is used. kOhm is the unit of an impedance.

Finally, the quality of the English needs to be improved.

Reviewer 2 Report

In this work, authors fabricated an impedance cytometry with the ITO, and employed it to test the cell viability. I reject this work because it does not seem well-prepared for publication

Comments:

1. The experiments were too simple and did not provide any new findings. Numerous works have verified the capability of impedance cytometry in detecting cell viability. Besides, my suggestion is to co-flow dead and live cells to show the sensitivity of the system. Separate measurements are not enough to convince me.

2. Page 2, Line 86. This statement is not supported by experimental results: "100-micron indium tin oxide (ITO) electrodes are comparable to 10-micron Au electrodes." BTW, most Au electrodes are nano scale in thickness. 

3. ITO electrode cannot be used a novel points, which has been employed in several publications, e.g.:

"Zhu, Shu, et al. "An easy-fabricated and disposable polymer-film microfluidic impedance cytometer for cell sensing." Analytica Chimica Acta 1175 (2021): 338759."

Reviewer 3 Report

In this manuscript, the authors developed a low-cost microfluidic device based on impedance cytometry for cell viability detection. The authors utilized 100um ITO electrodes to achieve similar performance with Au/Cr electrodes with finer technology node which is quite impressive. Some of the characterization of size dependent study and cell viability measurements are also included in the manuscript. Overall, this study is well-organized and provide a potential solution for mass applications. I have the following comments and suggestions that hope the authors can help address:

1. In the abstract part, the authors mentioned the proposed device performance is comparable to 10 um gold electrode. It would be great to include a comparison table for the proposed technology and existing technology to show the performance metrics are comparable.

2. In the abstract part, the authors mentioned the device cost can be as low as $1. It would be great to cover the cost breakdown for the readers to understand since the biggest selling point for this technology is cost.

3. In the discussion and conclusions, it would be great for the authors to discuss how this technology can be massive produced. It will bring more interests from the readers if this technology is cheaper and can be easily produced massively.

4. If possible, could the authors discuss a little bit on the performance consistency between different devices. That means if strict calibration is needed upon produced.

5. Even the authors demonstrate from the data they can identify dead and living cells, I still believe flow cytometry needs to be integrated to separate those two.  Maybe that could be included in the discussion part.

Reviewer 4 Report

This manuscript aims to assess the performance of their in-house developed microfluidic device for label-free sensing of cell viability. The authors conclude that the device has 94.5% accuracy in cell sorting and has comparable performance to 10um ITO electrodes MIC device. I have the following major comments about the manuscript, which I have also highlighted as comments in the attached PDF.

1. It is unclear how 94.5% accuracy was calculated in this manuscript. How does the ratio of number of live to dead cells calculated from the histogram in Fig. 6B compare with the ratio of live to dead cells in the input sample?

2. The authors claim comparable performance with 10um ITO electrodes but their is no data to support this claim. Could the authors compare how their method compares to FACS or 10um ITO electrode MIC in terms of accuracy of calling live and dead cells?

3. In the bead experiment, how did the bead ratio after counting compare to the bead ration for two sizes in input sample?

4. The authors claim 1$ per device as the cost of fabrication. How was this number calculated?

5. Is the difference in opacity statistically significant between live and dead cells?

6. The English language used in the manuscript can be improved overall to improve the accessibility of the paper to a worldwide audience. I suggest you have a colleague who is proficient in English and familiar with the subject matter, review your manuscript, or contact a professional editing service. Particularly, it would help to have the Methods section be in passive voice.

7. Could the authors clarify how cell alignment and ordering was achieved with the use of polyethylene oxide? Generally, the width of the channel is 100um which means multiple cells will be landing on the detection region at the same time. If that's the case, how do the authors de-convolute the impedance signal to understand its relation with a given individual cell?

8. I have other minor comments that are highlighted in the attached PDF.

Round 2

Reviewer 2 Report

All questions have been addressed. No more comments. 

Author Response

Dear Reviewer,

Thanks a lot for your comments.

Sincerely,

Jie Jayne Wu

Reviewer 4 Report

The authors have responded to my comments but I don't see appropriate changes in the manuscript.

1. For 94.5% accuracy, the revised manuscript still doesn't show the FACS data used to calculate the accuracy. The authors are using FACS data as the ground truth but it is not shown in the manuscript. Could the authors add a figure to the manuscript showing the FACS data from the test sample so that the readers can draw their own conclusions. Also, the methods section is missing details about FACS used the test sample. 

2. To my comment on the bead experiment, I enquired about how the bead ratio after counting from MIC compares to the bead ration for two sizes in input sample, and the authors responded that it is "basically the same". Again, could the authors provide the data from Coulter counter in the manuscript and quantify what they mean by "basically the same"?

3. To my original comment on whether the difference in opacity is statistically significant between live and dead cells, the authors mention that there is a clear distinction, but could the authors use an appropriate statistical test to confirm if the difference in opacity is significant or not?

Response 3: Thank you for your comments. Our device has high accuracy in particle counting. Before the beads injection into our device, a Coulter counter was used to count the beads. And the ratio of the numbers of beads at the output is basically in line with the ratio of the input beads. 
